# Mitochondrial Dynamics and Insulin Secretion

**DOI:** 10.3390/ijms241813782

**Published:** 2023-09-07

**Authors:** Uma D. Kabra, Martin Jastroch

**Affiliations:** 1Department of Pharmaceutical Chemistry, Parul Institute of Pharmacy, Parul University, Vadodara 391760, India; uma.kabra16205@paruluniversity.ac.in; 2The Arrhenius Laboratories F3, Department of Molecular Biosciences, The Wenner-Gren Institute, Stockholm University, SE-106 91 Stockholm, Sweden

**Keywords:** diabetes, mitochondrial dynamics, glucose-stimulated insulin secretion, pancreatic beta cell, fusion, fission

## Abstract

Mitochondria are involved in the regulation of cellular energy metabolism, calcium homeostasis, and apoptosis. For mitochondrial quality control, dynamic processes, such as mitochondrial fission and fusion, are necessary to maintain shape and function. Disturbances of mitochondrial dynamics lead to dysfunctional mitochondria, which contribute to the development and progression of numerous diseases, including Type 2 Diabetes (T2D). Compelling evidence has been put forward that mitochondrial dynamics play a significant role in the metabolism-secretion coupling of pancreatic β cells. The disruption of mitochondrial dynamics is linked to defects in energy production and increased apoptosis, ultimately impairing insulin secretion and β cell death. This review provides an overview of molecular mechanisms controlling mitochondrial dynamics, their dysfunction in pancreatic β cells, and pharmaceutical agents targeting mitochondrial dynamic proteins, such as mitochondrial division inhibitor-1 (mdivi-1), dynasore, P110, and 15-oxospiramilactone (S3).

## 1. Introduction: Diabetes—A Global Epidemic

Global diabetes mellitus incidences are increasing tremendously in all age groups, affecting the well-being and quality of life of many individuals [1]. Reports from the International Diabetes Federation stated that nearly 537 million people are suffering from diabetes, which is anticipated to increase to 643 million by 2030 and 783 million by 2045 [2]. Diabetes is a multifactorial disease, usually associated with persistently high glucose levels (hyperglycemia), either due to impaired insulin secretion or action [3]. It is categorized into two distinct types: Type 1 (T1D) and Type 2 Diabetes (T2D), wherein T2D accounts for at least 90–95% of total diabetic cases. In people with T1D, also known as early-childhood or juvenile diabetes, the immune system obliterates insulin-producing pancreatic beta (β) cells. In contrast, T2D usually develops later, and the patients are characterized by a blend of two metabolic dysfunctions: insulin resistance and inadequate insulin secretion [3,4]. Under conditions of insulin resistance, where insulin-dependent glucose uptake in peripheral organs is reduced, the β cells produce more insulin to compensate, leading to hyperinsulinemia [5]. Moreover, in some individuals, genetically compromised cells fail to secrete an adequate amount of insulin, resulting in hyperglycemia, the hallmark of T2D. Ultimately, cell mass and functions decrease, further aggravating the pathology [6]. Whether the exhaustion of β cells is caused by elevated insulin production [7] or dedifferentiation due to other metabolic complications [8] remains to be determined. Epidemiological studies reveal that age, lifestyle, ethnicity, smoking, and obesity also contribute to T2D [4]. Diabetes co-morbidities such as cardiovascular diseases, peripheral vascular diseases, neuropathy, retinopathy, stroke, and nephropathy are the main reasons for mortality in T2D individuals [9,10]. Despite extensive research, the pathophysiology related to the progression and complications of T2D is not fully understood. Definitely, to comprehend the underlying mechanisms associated with the disease, it is required to understand the concept of glucose homeostasis (glycemia).

### 1.1. Glucose Homeostasis

All mammalian cells require glucose as a metabolic substrate for energy production. Thus, it is necessary to sustain adequate levels of blood sugar in the range of 4–7 mmol/L. To maintain normal blood glucose levels, various hormones are released from the brain, liver, adipose tissues, pancreas, intestine, and muscles [3]. Among these, the endocrine pancreas plays a very critical role in regulating fuel storage by secreting several hormones. Mature endocrine cells, constituting only 2–3% of the total pancreatic volume, aggregate to form a discrete group of cells, which are termed the islets of Langerhans. These pancreatic islets consist of five types of endocrine cells: alpha-cells (15–20%) producing glucagon, beta-cells (65–80%) producing insulin and C-peptide, delta-cells (3–10%) producing somatostatin, gamma cells (3–5%) producing pancreatic polypeptide (PP), and epsilon-cells (<1%) producing ghrelin [11]. Human islets display a unique architecture wherein all the endocrine cells are bordered by the blood vessels, β cells are mostly located in the center, and α cells occupy the mantle position of the islet. As a result, the ratio of β cells to α cells is relatively higher in the core as compared to the mantle part of the islets [12]. The highly specialized blood supply pattern through the islet of Langerhans allows the ready exchange of molecules. The mass of β cells is maintained by the differentiation and replication of existing β cells, which are governed by different cell cycle machinery [13]. The islet cell organization of the diabetic patient and the non-diabetic patient does not differ significantly. The quantitative analysis, however, demonstrated a substantial reduction in β cell mass relative to α cell mass in type 2 diabetic subjects [14]. Two antagonistic hormones, insulin and glucagon, are crucial in sustaining blood sugar levels in the body [15]. In response to low blood glucose levels (hypoglycemia), glucagon promotes hepatic glucose release by glycogen breakdown and gluconeogenesis and enhances lipolysis in adipose tissue. In contrast, insulin has a counter-regulatory effect by reducing high blood glucose levels (hyperglycemia). Insulin stimulates glycogen production (glycogenesis) and glucose uptake in the skeletal muscle, liver, and adipose tissues, respectively. Alongside, hepatic glucogenesis and glycogenolysis are decreased, and lipolysis is potently inhibited [15,16]. To sense the blood glucose level for regulation of glucose homeostasis, glucose is transported to and metabolized in cells proportionally to the extracellular level [15,16,17].

### 1.2. Glucose-Stimulated Insulin Secretion

Glucose-Stimulated Insulin Secretion (GSIS) is a complex event modulated by the integration and interactions of multiple signal transductions in β cells [18]. Although glucose is the chief stimulator of insulin secretion, several hormones, nutrients, neural inputs, chemical messengers, and drugs modulate insulin release [18,19]. Following a meal, elevated blood glucose levels lead to glucose uptake via facilitated glucose transport into pancreatic β cells. In the cytoplasm, glucose is metabolized through glycolysis, thereby generating adenosine triphosphate (ATP), pyruvate, nicotinamide adenine dinucleotide (NADH), and water molecules [20,21]. The formed pyruvate enters the mitochondrial matrix, and is converted into acetyl-Coenzyme A (CoA), which enters the tricarboxylic acid (TCA) cycle for the generation of the reducing equivalents nicotinamide adenine dinucleotide (NADH) and flavin adenine dinucleotide (FADH2), guanosine triphosphate (GTP), and carbon dioxide (CO_2_). The reduced electron carriers NADH and FADH transfer their high-energy electrons to the electron transport chain (ETC), which harvests the energy by step-wise transport to drive protons into the intermembrane space, forming a proton gradient. The proton gradient is used to make ATP by driving the ATP synthase, a process known as oxidative phosphorylation (OXPHOS) [21,22,23]. Specifically, in β cells, the generated ATP elevates the cytoplasmic ATP/ADP ratio, which in turn closes the ATP-sensitive potassium channel, resulting in plasma membrane depolarization. This initiates the opening of L-type voltage-gated calcium channels, facilitating calcium (Ca^2+^) ions’ influx into the cells and thereby triggering the exocytosis of insulin granules [24,25]. From the above-described sequence of events, it is evident that the mitochondria of pancreatic β cells control GSIS, in particular by transferring the stored energy of glucose to ATP, the main trigger of insulin secretion, as represented in Figure 1.

GSIS is biphasic and has been considered a combined effect of both triggering and amplifying pathways, as shown in Figure 1. However, the amplifying pathways are more diverse and detailed as compared to the well-defined triggering pathways [26]. The triggering pathway initiates the first phase of insulin secretion lasting for 5–10 mins by the K_ATP_-dependent mechanism, whereas the second phase of insulin secretion lasts for hours, and the majority of insulin is released in this phase. A sustained second phase is enabled by the amplification pathway, which augments GSIS by K_ATP_-independent metabolic signals generated by byproducts of glucose metabolism such as nicotinamide, adenine dinucleotide phosphate hydrogen (NADPH), guanosine triphosphate (GTP), glutamate, and malonyl-CoA [27]. Several hormones and neurotransmitters acting on membrane receptors regulate the amplification of insulin secretion, such as cyclic adenosine monophosphate (cAMP), diacylglycerol (DAG), and plasma membrane phosphoinositides. In addition to metabolic signals, lipid metabolism also controls and modifies insulin secretion. Glucose-stimulated activation of cell division cycle 42 (Cdc42) pathways is important for second-phase insulin secretion, as it controls the mobilization and exocytosis of insulin. Reports suggest the impairment of amplifying pathways in β cells in both type 2 diabetic animal models and patients. Drugs that can rectify amplifying pathways, e.g., Glucagon-like Peptide 1 (GLP-1) conjugates, could be useful in overriding β cell dysfunction and enhancing insulin secretion in T2D [26,27,28].

## 2. Mitochondrial Dynamics in Diabetes

Mitochondria are organelles in eukaryotic cells, comprising an inner and an outer membrane separated by the intermembrane space. Beyond their primary role in fueling energy metabolism, they are also involved in several processes including cell signaling, calcium homeostasis, and apoptosis [29]. The crucial role of mitochondria in metabolic disorders, such as diabetes, is evident from the fact that in β cells, around 80% of glucose oxidation takes place in mitochondria. Therefore, hampering the mitochondrial energy metabolism by blocking the ETC, for example, impairs GSIS [30]. The significance of mitochondria in metabolism-secretion coupling has further been illustrated in Rho cells devoid of mitochondrial DNA (mtDNA) that no longer respond to glucose. These cells exhibit higher NAD(P)H levels, which in turn inhibit the enzyme glyceraldehyde phosphate dehydrogenase but enhance lactate production by the lactate dehydrogenase enzyme [31]. The above findings highlight the significance of hydrogen shuttles and mitochondrial respiration in re-oxidizing NAD formed during glycolysis in β cells. The effect of mitochondrial dysfunction is also demonstrated by mtDNA mutations, e.g., the A3243G mutation in a transfer RNA (tRNA), which causes a significant decline in both first- and second-phase insulin secretion [32]. Further, the knockout of mitochondrial transcription factors A (Tfam) in β cells impair OXPHOS and GSIS, resulting in diabetic mice [33]. The mutation m.8561C>G in the subunit of mitochondrial ATP synthase (*MT*-*ATP6*/*8)* resulted in impaired complex assembly and decreased ATP production, causing peripheral neuropathy and diabetes mellitus [34]. The above findings clearly demonstrate the critical role of mitochondria and their function in insulin secretion. Importantly, mitochondria exist as a dynamic reticular network that frequently undergoes repetitive cycles of fission and fusion in a regulated manner, referred to as “mitochondrial dynamics.” The mitodynamics are regulated by a group of highly conserved dynamin-related GTPases [29] as shown in Figure 2. In humans, mice, and rats, β cell mitochondria exist as densely interconnected tubules throughout the cytoplasm [35]. In the β cell, mitochondrial dynamics compensate for damaged and dysfunctional mitochondria by fusing them with functional ones. On the other hand, fission can drive the removal of damaged or non-functional mitochondria through mitophagy [36]. It is conceivable to state that a counterbalance between the two dynamic processes is required for normal mitochondrial functionality. Accumulated pieces of evidence support the notion that disturbances in the tuning of the fusion/fission process in the pancreatic β cell led to the development and progression of diabetes in animal and human models.

Chronic nutrient exposure (e.g., glucose or free fatty acids) has been associated with increased ROS production, mitochondrial dysfunction, and enhanced cell apoptosis, thereby perpetuating diabetes and its complications [37,38,39]. A previous study demonstrated that palmitate-induced mitochondrial fragmentation increased ROS production, lowered ATP production, and increased cell apoptosis [40]. Similarly, in muscle cells, palmitate treatment caused lipid accumulation, increased oxidative stress, induced mitochondrial fission, and increased insulin resistance [41]. In prediabetic Zucker diabetic fatty rats, Troglitazone (TZD) treatment protected β cells from lipotoxicity and lipo-apoptosis by enhancing the activity of plasma lipoprotein lipase (LPL), thereby lowering the triglyceride content. TZDs also prevented mitochondrial alteration, improved insulin sensitivity, and henceforth impaired GSIS [42]. Similarly, β cell mitochondria obtained from diabetic GotoKakizaki rats were found to be disconnected, swollen, and shorter [43], indicating possible disruption of mitochondrial structure. Islets obtained from type 2 diabetic patients showed disrupted mitochondrial structure, reduced ATP levels, and decreased amounts of insulin granules, with a consequent reduction in insulin secretion [44,45]. In INS 1e cells and in islets, hyperglycemia-induced mitochondrial fragmentation decreased membrane potential, increased ROS production, and eventually promoted β cell apoptosis [46]. Given the evidence for the structure-function relationship of mitochondria, mitochondrial dynamics-regulating proteins are certainly involved in the pathophysiology of diabetes.

In the present review, we discuss the functional role of mitochondrial dynamics as a significant contributing factor to the perpetuation of normal pancreatic β cell function. The underlying mechanisms have been addressed by altering the expression of mitochondrial dynamics proteins using pharmacological and genetic tools, the latter either by overexpression or knockdown.

### 2.1. Mitochondrial Fusion and Its Machinery

Mitochondrial fusion is considered the merging of the outer and inner mitochondrial membranes of two different mitochondria to form a larger unit. Fusion enables the rapid exchange of metabolites between neighboring mitochondria, thereby complementing impaired mitochondria and promoting their functionality [36]. Mitochondrial fusion is also required for the maintenance and distribution of mtDNA. Before entering the S phase of the cell cycle, hyperfusion of mitochondria takes place, thereby increasing ATP production [36,47]. The three large dynamin-family GTPases responsible for mitochondrial fusion in mammals are Mitofusins 1 and 2 (Mfn1 and 2) and optic atrophy 1 (Opa1), respectively [48] as depicted in Figure 2.

#### 2.1.1. Outer Membrane Fusion Proteins: Mitofusins (Mfn1/2)

Mitofusins 1 and 2 are embedded in the outer mitochondrial membrane (OMM) and tether mitochondria together by forming complexes in the trans conformation [47,48]. Both proteins share structural homology but are functionally different. Mfn1 shows greater GTPase activity and more efficient fusion relative to Mfn2 [49]. However, Mfn1 and Mfn2 can functionally substitute each other, and the lack of both mitofusins eliminates mitochondrial fusion. For instance, in fibroblasts, deletion of both Mfn1 and Mfn2 resulted in complete mitigation of mitochondrial fusion, and the fibroblasts exhibited poor growth, decreased mitochondrial membrane potential, and reduced respiration [49,50]. Genetic deletion of either mitofusin causes mitochondrial dysfunction and embryonic lethality [50]. For example, muscle cells lacking Mfn2 displayed a decrease in glucose oxidation, mitochondrial membrane potential, and mitochondrial respiration [51]. Overexpression of Mfn1T109A, a point mutation in the GTPase domain of Mfn1, acts as a dominant-negative (DN) and results in extreme fission [52]. Missense mutations in Mfn2 are responsible for Charcot-Marie-tooth disorder type 2A2 (CMT2A2), a neurodegenerative disease [53]. Overexpression of wild-type Mfn1 (WT-Mfn1) shifts mitochondria towards fusion, resulting in perinuclear mitochondrial aggregation in the pancreatic INS-1e cell culture model and primary β cells. This leads to increased lactate production in INS1e cells, shunting pyruvate away from complete oxidation in mitochondria, and decreased cellular ATP levels at both basal and stimulatory glucose concentrations, causing impaired GSIS [54]. In a similar study, overexpression of WT-Mfn1 induced hypomotility and impaired functionality of mitochondria, which were attributed to Mfn1-induced mitochondrial aggregation, restricting the entrance into the subplasma membrane area and reducing mitophagy [55]. Adding further complexity to our understanding of mitodynamics in β cells, overexpression of the DN-Mfn1 gene results in small, discrete mitochondria extensively distributed in the periphery due to disruption of mitochondria in the microtubule system. However, no statistically significant changes occurred in apoptosis, mitochondrial hyperpolarization, or metabolism-secretion coupling [54,55]. Hence, mitochondrial fragmentation as such does not seem to affect insulin secretion, at least not in vitro. In a recent study, Mfn1 or Mfn2 β cell-specific knockout mice exhibited normal glucose homeostasis with no changes in insulin secretion. In contrast, double knockout mice with β cell-specific Mfn1 and 2 deletions displayed glucose intolerance and impaired insulin secretion due to loss of mtDNA content. Interestingly, Tfam overexpression in fusion-deficient β cells ameliorated the reduced mtDNA copy number but impaired GSIS. Furthermore, the pharmacologic agonism of mitofusin was capable of rescuing mtDNA content and GSIS in islets from db/db mice, a model of T2D [56]. Consistently, the pancreatic β cell-specific knockout of Mfn1/2 in mice (βMfn1/2 KO) led to several metabolic abnormalities, such as glucose intolerance, impaired glucose clearance, and a decrease in plasma insulin levels. βMfn1/2 KO mice displayed a higher degree of fragmented mitochondria and disrupted cristae structure. Under hyperglycemic conditions, the KO mice showed a reduction of mitochondrial calcium accumulation and hyperpolarization, associated with impaired GSIS [57,58] which could be rescued by treatment with glucagon-like peptide 1 (GLP1) or glucose-dependent insulinotropic peptide receptor (GIP) [58,59]. Indirect evidence was observed in Mfn1 KO mice, which displayed defective mitochondrial structure and flexibility in pro-opiomelanocortin (POMC) neurons and defective insulin secretion by pancreatic β cells [60].

#### 2.1.2. Inner Membrane Fusion Protein: Optic Atrophy 1 (Opa1)

Opa1 is positioned in the inner mitochondrial membrane (IMM) through the intermembrane space of mitochondria [47,61]. It also promotes Mfn1-mediated mitochondrial fusion of the outer membrane [48,62]. The defect in Opa1 causes autosomal dominant optic atrophy, an inherited optic neuropathy condition [63]. As suggested by several in vitro cell culture studies, Opa1 plays a vital role in the intrinsic apoptotic pathway and the maintenance of mtDNA [64,65,66,67,68]. Possibly due to the loss of mtDNA, the reduction in mitochondrial fusion results in decreased oxidative phosphorylation [47,68]. In RIP2-Opa 1 KO islets, decreased levels and activity of electron transport chain complex IV resulted in impaired oxygen consumption rate, calcium signaling, and insulin secretion. The mice were found to be hyperglycemic. Notably, however, the total amount of mtDNA remained unchanged in this case. In vivo, studies further confirmed that β cell-specific Opa1 knockout mice are hyperglycemic and exhibit dismantled mitochondria with abnormal cristae structures. Additionally, these mice displayed reduced β cell proliferation, decreased ATP production, and GSIS [69]. Mild overexpression of Opa1 in INS1e cells led to more elongated and tubular mitochondria, whereas higher levels of Opa1 overexpression resulted in an increased number of fragmented mitochondria [70,71].

### 2.2. Mitochondrial Fission and Its Machinery

Mitochondrial fission is defined as the division of mitochondria into two new organelles. Fission is necessary for growing cells to provide them with a sufficient number of mitochondria [47,48]. During cellular stress conditions, fission enables the removal of damaged mitochondria and promotes apoptosis, thus helping quality control [61,72,73]. Outer mitochondrial fission is mediated by conserved dynamin family GTPases, mitochondrial fission 1 protein (Fis1), and dynamin-related protein 1 (Drp1) [48] as shown in Figure 2.

#### 2.2.1. Mitochondrial Fission 1 Protein (Fis1)

Fis1 induces fission by different mechanisms in eukaryotes. In yeast, Fis1 executes mitochondrial fragmentation by recruiting the Drp1 homolog Dnm1p to the mitochondrial outer membrane (OMM) [48,61]. However, in humans, hFis1 promotes mitochondrial fragmentation by two different mechanisms: firstly, it is thought to have a similar role as the homolog Dnm1p, i.e., the mitochondrial recruitment of Drp1 and the initiation of fragmentation; secondly, it binds to pro-fusion proteins Mfn1, Mfn2, and Opa1, and blocks mitochondrial fusion by inhibiting the GTPase activity [47,74,75]. In addition, Fis1 has also been involved in apoptotic and autophagic pathways [76]. In primary pancreatic β cells and INS-1e cells, the alteration of fission by silencing Fis1 resulted in compromised mitophagy, accounting for the accumulation of dysfunctional mitochondria, reduced respiratory function, and GSIS. However, the Fis1 knockdown did not significantly alter mitochondrial morphology [71,77]. Intriguingly, in the reverse experiment of Fis1 overexpression, mitochondrial fragmentation was remarkably induced and resulted in a similar phenotype of bioenergetics dysfunction, i.e., increased lactate production, impaired glucose-induced hyperpolarization, and reduced ATP levels, accompanied by decreased cytosolic and mitochondrial calcium release, all accounting for impaired insulin secretion [54,77]. Schultz et al. reported reduced Fis1 expression in glucose-unresponsive cells INS832/2 vs. responsive cells INS832/13, along with a greater number of elongated mitochondria in INS832/2 cells. Interestingly, lentiviral overexpression of Fis1 in INS832/2 cells induced a more homogenous mitochondrial network and enhanced insulin secretion. On the contrary, silencing of Fis1 in INS832/13 cells and primary mouse β cells increased mitochondrial elongation, decreasing GSIS. However, the expression levels of electron transport chain complexes and ATP synthase were unaffected. Further overexpression of Fis1 in primary mouse β cells reduced insulin secretion and fragmented mitochondria. In contrast, Fis1 overexpression improved insulin secretion in INS832/13 cells. Furthermore, a stepwise increase of Fis1 levels in the less glucose-responsive insulin-producing cell line RINm5F of the rat resulted in improved GSIS, whereas high overexpression was adverse, resulting in mitochondrial clusters and diminished GSIS [78]. The above finding confirms the significant role of Fis1 in regulating metabolism-secretion coupling in pancreatic β cells, although the effect depends on the expression levels of Fis1. However, Fis1 RNAi maintained mitochondrial dynamics by favoring fusion and preventing apoptosis [71].

#### 2.2.2. Dynamin Related Protein 1 (Drp1)

In mammals, mitochondrial fragmentation is mainly mediated by a highly conserved GTPase protein, namely dynamin-related protein 1 (Drp1). Drp1 is distinctly expressed in various tissues; high levels are expressed in the brain, muscle, and endocrine tissues; moderate levels are found in the kidney, lung, pancreas, and liver, and low levels are detected in the ovaries [79,80]. There are four domains in Drp1: (1) the N-terminus GTPase domain, which forms a dimer, stabilizes the active sites, and stimulates GTPase activity; (2) the variable domain, which contains most of the post-translational modification sites; (3) the helical middle assembly domain, which promotes Drp1 self-assembly into higher-order structures; and (4) the C-terminus GTPase effector domain, which mediates both intra and intermolecular interactions, as represented in Figure 3. However, Drp1 lacks a C-terminus proline-rich domain and pleckstrin homology domain [80,81]. Drp1 consists of 21 exons, and alternate splicing of exons 3, 16, and 17 gives rise to multiple isoforms with differential GTPase activity [82,83]. Different isoforms are expressed differently in various tissues. The longer Drp1 isoform is expressed predominantly in neurons and consists of distinctive polypeptide sequences within their GTPase and variable domain, called A-insert (encode for exon 3) and B-insert (encode for exons 16 and 17), respectively [82,83,84]. The widely expressed and shorter isoforms of Drp1 lack the A insert and alternatively exclude either exon 16, 17, or both and differentially regulate the geometry and curvature of Drp1 on the fission sites [82,83]. Drp1 mostly resides in the cytoplasm, and about 3% of the total protein dwells at the mitochondrial surface. Several cellular stimuli, such as Ca^2+^ concentration and apoptosis, activate the recruitment of Drp1 to the OMM and self-associate with adapter proteins, mitochondrial fission factor (Mff), fission protein 1 (Fis1), and mitochondrial dynamics proteins of 49 and 51 KDa (Mid49 and Mid51) [48,75]. After association, Drp1 forms a higher-order assembly on prospective OMM fission sites, followed by GTP hydrolysis, causing conformational changes and inducing mitochondrial fission [85]. The incorporation of the dominant negative (DN) mutant of Drp1 K38A inhibits membrane constriction and blocks organelle fission [86]. In mammalian cells, inhibition of Mff or double knockdown of MiD49 and MiD51 decreases Drp1 translocation to mitochondria and promotes elongation [87,88]. It has also been reported that MiD49 and MiD51 proteins are capable of controlling mitochondrial fission independently of Fis1 and Mff [89]. Apart from regulating mitochondrial fission, Drp1 is also thought to be involved in mediating vesicle formation, endoplasmic reticulum morphology, and peroxisomal fission in mammals [90]. The function of the fission gene Drp1 is influenced by various posttranslational modifications such as phosphorylation, sumoylation, ubiquitination, and S-nitrosylation. These posttranslational modifications; along with protein effectors as shown in Figure 3, are known to modulate the stability, localization, and GTPase activity of Drp1 in various physiological and pathological conditions [91]. Drp1 phosphorylation by the Cdk1/Cyclin B complex at serine 616 is necessary to trigger mitochondrial fission in mitotic cells to allow even mitochondrial distribution to progeny [92]. An increased level of phosphorylation at the Ser 616 site has been observed in Alzheimer’s patients [93]. The protein kinase (PKA)-dependent Drp1 phosphorylation at Ser 616 enhances mitochondrial fission, leading to hypertensive encephalopathy [94]. The cyclic adenosine monophosphate (cAMP)-mediated PKA-dependent Drp1 phosphorylation at Ser 637 attenuates Drp1 GTPase activity along with intermolecular interaction, resulting in reduced mitochondrial fission [95]. Activation of AMP-activated protein kinase (AMPK) increased Drp1 phosphorylation at anti-fission site Ser 637 and prevented the alteration in endoplasmic reticulum morphology, decreased mitochondrial fragmentation, and reduced apoptosis in energy-stressed β cell [96]. Elongated mitochondria are spared during autophagy and maintain ATP production and cell viability, specifically during starvation [97]. On the contrary, calcineurin-dependent Drp1 dephosphorylation at Ser 637 boosts its recruitment to mitochondria and favors mitochondrial fission [98]. SUMO ligases like Sumo1, Ubc9, and MAPL mediate Drp1 sumoylation at different lysine residues within the variable domain and exert an effect on its interaction with the OMM or other proteins [99]. Overexpression of Sumo 1 stabilizes Drp1 on mitochondria and prevents its degradation, thereby promoting mitochondrial fission [100]. SUMO protease SenP5 mediates the desumoylation of Drp1, which is essential for the elimination of SUMO-2/3 conjugates of Drp1 [101]. SUMO-specific protease 2 (SENP2) regulates Drp1 phosphorylation at the Ser 616 residue and insulin secretion in the NIT-1 pancreatic β cell line [102]. MARCH5-dependent K63-linked ubiquitination stabilizes Drp1 on mitochondria, whereas parkin-mediated K48-linked ubiquitination triggers the proteasomal degradation of Drp1 [103,104]. Drp1 can also be modified by nitrosylation at Cys644. In Alzheimer’s patients, S-nitrosylation of Drp1 at Cys644 promotes mitochondrial fission and neurotoxic events; however, preventing S-nitrosylation by the Cys644Ala mutation abrogated neurotoxicity [105]. In cardiomyocytes, O-linked-N-acetyl-glucosamine glycosylation (O-glcNAcylation) of Drp1 at residues T585 and T586 activates its recruitment to mitochondria and enhances fission [106]. The above findings certainly underscore the crucial role of Drp1 in regulating mitochondrial fission.

Global Drp1-knockout mice were found to be embryonic lethal due to the lack of mitochondrial fission [29]. Similarly, abnormal brain development was observed in newborn children with a heterozygous mutation in Drp1. Cells obtained from this patient displayed elongated and interconnected mitochondria [107]. All the above findings support the involvement of Drp1 in causing mitochondrial fission provoked by various cellular stimuli. Blocking of Drp1 function by RNAi or the DN allele gives rise to elongated and interconnected mitochondria that result in the degradation of mitochondrial mass. For instance, in HeLa cells, the knockdown of Drp1 induced a reduction in mtDNA and mitochondrial respiration [108]. Another study demonstrated that the downregulation of Drp1 prevented the decrease in mitochondrial membrane potential and the release of cytochrome c in COS-7 cells [71]. Whereas, in hippocampal neurons, loss of Drp1 function leads to misshaped synaptic vesicles [109]. The indispensable role of Drp1 in mitochondrial fragmentation was also recognized in the research field of diabetes. Huang et al. reported alterations in mitochondrial morphology and a decrease in ATP production due to the abnormal increase of Drp1 expression in a mouse model of T2D [110]. Several knockdown and chemical inhibition studies in β cell lines and pancreatic islets have emphasized the role of Drp1-dependent mitochondrial fission in the regulation of insulin secretion. Glucose stimulation of INS1e cells induced reversible shortening of mitochondria and promoted insulin secretion. However, the suppression of the fission event by the dominant negative (DN) mutant DLP1-K38A eliminated glucose-induced morphological changes, increased proton leak, decreased ATP production, and consequently GSIS [111]. In a similar study by Twig et al., inhibition of Drp1 by the DN mutant prevented mitochondrial autophagy, increased the accumulation of oxidized mitochondrial protein, and led to defective insulin secretion [71]. Further, the inhibition of fission by small hairpin Drp1 RNAs in INS1e cells and islets reduced the expression of mitochondrial fusion proteins, thereby shifting the mitochondrial morphology from moderate clusters to the elongated form. These morphological changes were responsible for reduced mitochondrial membrane potential, ATP production, and GSIS [112]. In the NIT1 pancreatic β cell line, Drp1 knockdown caused impairment of GSIS, which was restored by SENP 2 overexpression [102]. In our previous work, we have demonstrated that the substrate supply upstream of the oxidative phosphorylation machinery is hampered by the pharmacologic silencing of Drp1 in MIN6 cells and mouse pancreatic islets [113]. Recently, Bordt et al. suggested off-target effects of the Drp1 inhibitor mdivi-1 (mitochondrial division inhibitor-1) on complex I of the ETC [114]. However, genetic silencing of Drp1 achieved similar results as pharmacology. The direct supply of exogenous pyruvate fully rescued the deficiency in oxidative phosphorylation, ATP levels, and GSIS [113], strongly suggesting that Drp1 silencing affects mainly substrate supply. While increasing Drp1 expression would have been a feasible route to improve insulin secretion, transient Drp1 overexpression failed to rescue GSIS in Drp1-KD MIN6 cells, which was due to drastically reducing insulin content [115]. The effect of inhibiting or blocking mitochondrial fission has also been interrogated in vivo by generating β cell-specific Drp1 knockout mice (β Drp1KO). The islets exhibited highly fused mitochondrial morphology and impaired second-phase insulin secretion with no alteration in oxygen consumption rates or calcium ion influx [116]. Comprehensively, in pancreatic β cell lines and islets, genetic or pharmacologic silencing of Drp1 caused impairment in insulin secretion due to decreased ATP-linked respiration and/or increased mitochondrial proton leak, whereas in Drp1β KO islets, oxygen consumption remained unchanged. These discrepancies in past data may be attributed to the difference in proliferative β cell lines and dormant mouse islets or the effect of chronic vs. acute exposure. However, suppression of Drp1 activity ameliorated free fatty acid (FFA)-induced mitochondrial fragmentation, insulin resistance, and apoptosis in pancreatic cells and islets, as well as in muscle cells [40,41]. The effect of changed mitochondrial dynamic proteins on insulin secretion has been summarized in Figure 4. The genetic variation of mitochondrial dynamic proteins and their effect on mitochondrial morphology and β cell function are summarized in Table 1.

## 3. Targeting Mitochondrial Dynamic Proteins

Drp1 also plays a critical role in the regulation of mitophagy as shown in Figure 2, thus helping to maintain mitochondrial integrity and function necessary for cell survival [118,119,120,121]. Previous studies also demonstrated mitochondrial fission followed by selective fusion and elimination of dysfunctional mitochondria through mitophagy [71]. Accumulation of dysfunctional mitochondria and increased ROS production were reported in Type 2 diabetic patients [122,123,124]. In T2D, β cells are exposed to high glucose which is associated with increased oxidative stress and disrupted mitochondrial morphology that hinders the mitophagy pathway and related genes and eventually leads to increased insulin resistance and beta cell death [119,122]. Twig et al demonstrated that inhibition of mitochondrial fission by DRP1^K38A^or FIS1 RNAi resulted in decreased mitophagy, reduced mitochondrial respiration, and impaired insulin secretion [71,72]. The indispensable molecular players of the mitophagy process are PTEN-induced putative kinase 1 (PINK1), E3 ubiquitin ligase PARKIN and CLEC16A, cardiolipin, transcriptional regulators like Transcription factor B2 (TFB2M), and orphan nuclear receptor Nor1/NR4A3 [125,126,127,128,129]. Altered expression of these proteins alters mitochondrial dynamics and impairs the mitophagy process leading to the development of T2D [130]. The impairment of mitophagy can be restored by mitophagy inducers and can be used as a promising strategy for ameliorating T2D [119,131].

The proper functioning of β cell mitochondria is essential due to their involvement in insulin production and release. Unregulated mitochondrial dynamic protein function is associated with numerous pathologies, including T2D. Treatment strategies focusing on manipulating mitochondrial dynamics could possibly ameliorate β cell dysfunction and help maintain glucose homeostasis. Targeting regulators of mitochondrial dynamics in T2D would be a novel approach, but it is very challenging due to multifaceted molecular mechanisms that can lead to unexpected side effects. In recent years, different computational tools have aided in rational drug design and screening of therapeutically important small molecules. These advances have provided breakthroughs for the development of pharmacological compounds that can modulate mitochondrial dynamics. It is widely reported that the compound mdivi-1 inhibits the GTPase activity of Drp1, which is involved in mitochondrial fission. In C2C12 muscle cells, mdivi-1 treatment attenuated palmitic acid-induced mitochondrial fragmentation, oxidative stress, mitochondrial depolarization, and insulin resistance [41]. Delivery of mdivi-1 into diabetic mice reduced mitochondrial fission, ROS production, inflammation, atherosclerosis, and ameliorated endothelial function [132]. Mdivi-1 treatment exhibited a cardioprotective effect in HFD-STZ mice by reducing mitochondrial fission, improving mitochondrial function, and suppressing cardiomyocyte apoptosis [133]. However, a study in the pancreatic β cell line MIN6 and mouse islets demonstrated that mdivi-1 treatment impaired insulin secretion by affecting substrate supply upstream of mitochondria [113]. Mdivi-1 reversibly inhibits mitochondrial complex I-O_2_ consumption and ROS production independently of Drp1 [114]. Collectively, the short-term potential benefits of mdivi-1 are well demonstrated and can be used in the management of macrovascular complications in T2D. On the other hand, long-term treatment with mdivi-1 suppresses mitochondrial function, decreases mitochondrial mass, and promotes apoptosis in vascular smooth muscles [134]. Another compound that is a competitive inhibitor of the GTPase activity of Drp1 is Dynasore. It acts by inhibiting the endocytic pathway by blocking coated vesicle formation [135]. In ischemia/reperfusion mice, dynasore treatment prevented mitochondrial fragmentation and oxidative stress, thereby improving overall cardiac function [136]. P110, a small peptide inhibitor, blocks the Drp1/Fis1 interaction, which is necessary to dock Drp1 on mitochondria. In cultured neurons, it blocks the binding of Drp1 to Fis1 and exhibits a neuroprotective effect by inhibiting mitochondrial fragmentation and ROS production [137]. Furthermore, P110 prevented the association of Drp1 with p53 in the ischemia/reperfusion rat and decreased brain infarction and necrotic cell death [138]. Lastly, 15-Oxospiramilactone (S3), a diterpenoid derivative, inhibits USP30 (mitochondria localized-deubiquitinase) that promotes non-degradative ubiquitination of Mfn1/2, which further enhances Mfn1/2 activity and mitochondrial fusion. In *mfn1*^–/–^or *mfn2*^–/–^ knockout cells, it has been observed that S3 restored mitochondrial function and fusion [139]. In summary, mitochondrial fission inhibitors are promising therapeutic agents for diseases where increased Drp1 expression is involved in the pathology. It is important to note that limited data are available on their effects on metabolic diseases like T2D, and therefore further research is needed. Collectively, however, the application of fission-inhibitors, such as mdivi-1, Dynasore, and P110, or fusion-promoters, such as S3, is not obvious for the treatment of β cell dysfunction, where fission is required during at least the initial phase of GSIS.

## 4. Conclusions

T2D has emerged as one of the leading global health problems and is associated with insufficient insulin secretion from pancreatic β cells and peripheral insulin resistance. Mitochondria are highly dynamic organelles and play a vital role in maintaining energy homeostasis. Mitochondrial morphological changes upon glucose stimulation are necessary for proper insulin secretion. Accumulating pieces of evidence support the notion that mitochondrial fission and fusion cycles are essential for the metabolism-secretion coupling of pancreatic β cells both in vitro and in vivo. This review has provided an overview of the functional aspects of the alteration of mitochondrial dynamics protein on insulin secretion. Targeting mitochondrial dynamics is emerging as a potential therapeutic approach, including for T2D. In the review, we have also discussed compounds targeting mitochondrial dynamics, such as mdivi-1, dynasore, P110, and S3. However, the data are limited to cancer and brain diseases, and research on T2D still needs to be explored.

## Figures and Tables

**Figure 1 ijms-24-13782-f001:**
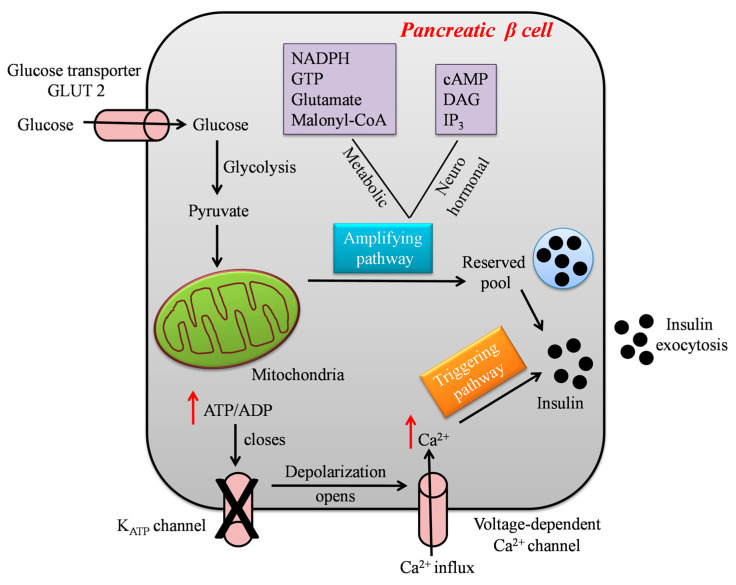
A schematic model showing the steps involved in glucose-stimulated insulin secretion by the pancreatic β cell.

**Figure 2 ijms-24-13782-f002:**
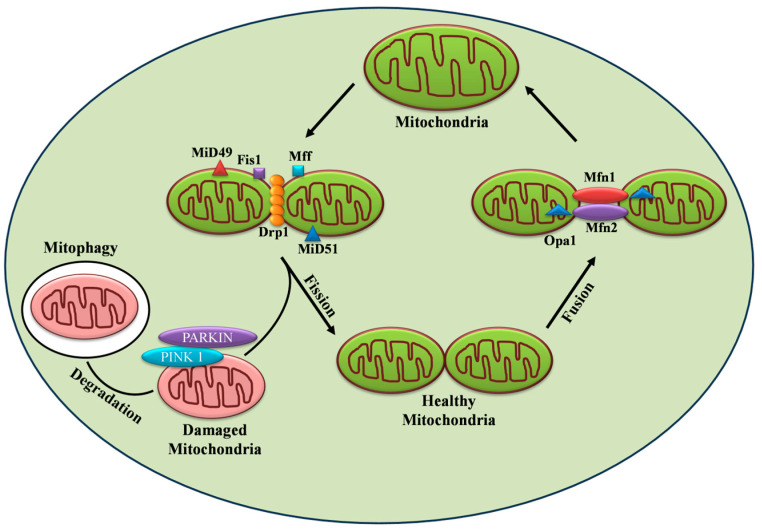
Regulation of mitochondrial dynamics by fission, fusion, and mitophagy.

**Figure 3 ijms-24-13782-f003:**
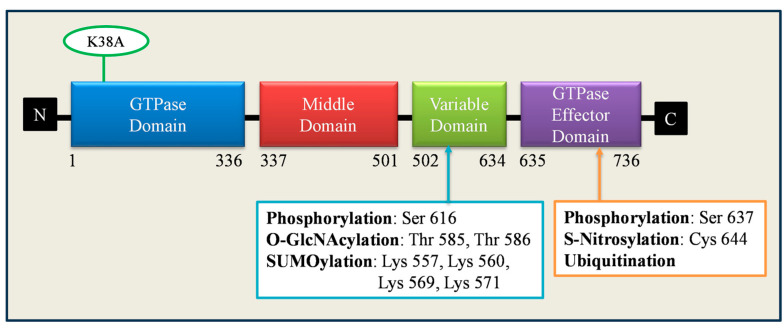
Domain structure and posttranslational modification of Drp1 at different amino acid residues, along with commonly used dominant-negative mutant (K38A).

**Figure 4 ijms-24-13782-f004:**
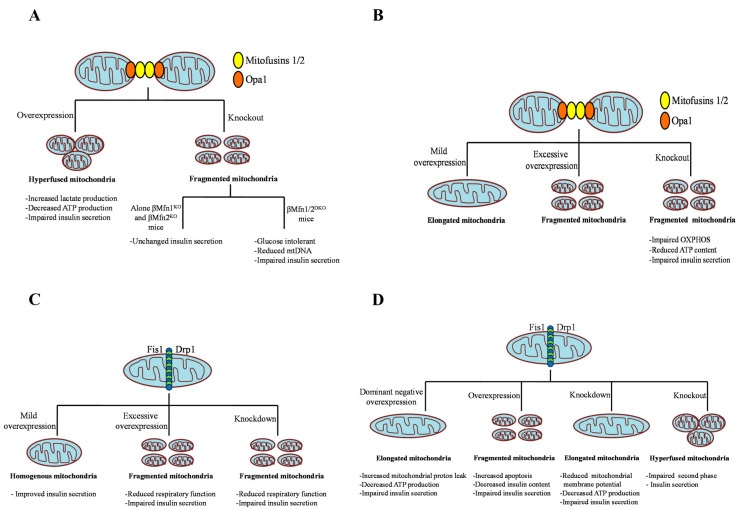
Effect of manipulation of mitochondrial dynamic regulating players on insulin secretion. (**A**) Mitofusins (Mfn1/2), (**B**) Optic Atrophy 1 (Opa1), (**C**) Mitochondrial Fission 1 Protein (Fis1), and (**D**) Dynamin Related Protein 1 (Drp1).

**Table 1 ijms-24-13782-t001:** Manipulation of mitochondrial dynamics proteins in pancreatic β cells and islets and their effect on mitochondrial morphology and β cell function.

Dynamin GTPase	Genetic Intervention	Effect on Mitochondrial Morphology	Effect on β Cell Function	References
Mfn1/2	Mfn1 overexpression in Ins1e cells and primary β cells	Hyperfused and aggregated	Increased lactate production, decreased cellular ATP levels, and impaired GSIS	[54]
	Mfn1 overexpression in Ins1e cells	Hyperfused and aggregated	Loss of mitophagy, hypomotility, and impaired mitochondrial function and insulin secretion	[55]
	DN-Mfn1 overexpression in INS1e cells	Discrete	No significant changes in apoptosis, mitochondrial hyperpolarization, and metabolism–secretion coupling	[61,62]
	βMfn1/2KO mice	Fragmented mitochondria, disrupted cristae shape and structures	Reduced Ca^2+^ accumulation, mitochondrial membrane potential, β cell connectivity, and GSIS	[57,58,59]
	Alone βMfn1^KO^ mice and βMfn2^KO^ mice	Fragmented	Normal glucose homeostasis and no change in insulin secretion	[56]
	βMfn1/2^DKO^ mice	Fragmented	Glucose intolerant, reduced mtDNA content, and reduced insulin secretion	[56]
Opa1	RIP2-Opa1 KO β cells	Fragmented mitochondria and abnormal cristae structures	Decreased complex IV levels, impaired OXPHOS, decreased ATP content, reduced insulin secretion and cell proliferation	[69]
	Mild overexpression in INS1e cell	Elongated	NA	[70,71]
	High overexpression in INS1e cell	Fragmented	NA	[70,71]
	Overexpression in primary β cell	Fragmented	NA	[70]
Fis1	Downregulation by RNAi in INS1e cells	Elongated	Reduced mitophagy, respiratory functions, and GSIS	[71]
	Knockdown by shRNA INS832/13 and primary mouse β cells	Elongated	Impaired GSIS but no changes in expression of OXPHOS complexes and ATP synthase activity	[66]
	Overexpression in INS 1e cells	Fragmented, reduced volume, and swollen mitochondria	Increased lactate production, reduced mitochondrial energy metabolism, and impaired insulin secretion	[54]
	Overexpression in primary mouse β cells	Fragmented	Reduced insulin secretion	[78]
	Overexpression in INS832/12 cells	Homogenous mitochondrial network	Enhanced insulin secretion	[78]
	Moderate overexpression in RINm5F cells	Homogenous mitochondrial network	Improved insulin secretion	[78]
	High overexpression in RINm5F cells	Clusters	Reduced insulin secretion	[78]
Drp1	Drp1 DN (DLP1-K38A) overexpression in INS1e cells	Hyperfused	No significant change in GSIS; prevented apoptosis	[117]
	Drp1 DN (DLP1-K38A) overexpression in INS1e cells	Swollen and elongated	Decreased mitochondrial autophagy, respiratory functions, and GSIS	[70,71]
	Drp1 DN (DLP1-K38A) overexpression in INS1e cells	Elongated	Increased mitochondrial proton leak, decreased ATP production, and GSIS	[111]
	Downregulation by shRNA in INS1e cells	Elongated	Decreased mitochondrial membrane potential, reduced ATP production, and GSIS	[112]
	Knockdown in NIT-1cells	Elongated	Impaired GSIS	[102]
	Knockdown genetically or pharmacological inactivation by Mdivi1 in MIN6 cells and islets	Elongated	Reduced mitochondrial ATP synthesis, and impaired GSIS due to compromised substrate delivery upstream of mitochondria	[113]
	β Drp1b-KO mice	Hyperfused	Normal oxygen consumption rate and calcium concentration but significantly impaired second-phase insulin secretion	[116]
	Overexpression in INS1e cells	Round and short	Increased cytochrome C release, and ROS production, no significant change in GSIS but tends towards decreased GSIS	[46,117]
	Overexpression in MIN6 cells	Fragmented	No effect on mitochondrial metabolism, impaired GSIS due to reduced insulin content	[115]

## Data Availability

Not applicable.

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
