# Peer review of "Mitochondrial Dynamics and Insulin Secretion"

_ijms, 2023, doi:10.3390/ijms241813782_

Round 1

Reviewer 1 Report

This is a well written review and provides a more comprehensive account of the involvement of mitochondrial dynamics in insulin secretion than previous  reviews related to the same topic. The review addresses the contribution of changes in mitochondrial dynamics and consequences on pancreatic beta cell insulin secretion and content directly. Clearly this review provides sufficient evidence to suggest an involvement of changes in mitochondrial fission or fusion events in insulin secretion by pancreatic beta cells, however it is unclear as to if this is a cause or consequence of the disease. 

Given that impairment of insulin secretion in diet-induced diabetes is related to elevated levels of circulating free fatty acids or lipid accumulation in islets. I wonder if the authors have any insight into effects of elevated fat levels and effects on mitochondrial dynamics and if they believe changes in dynamics to be involved in the causality of impaired insulin secretion or a consequence of other metabolic stresses that associate with type 2 diabetes. It would therefore be insightful to provide a section on any evidence linked to the pathophysiology of diabetes, mitochondrial dynamics and insulin secretion.

Since type-2 diabetes has multiple subtypes that are linked to different environmental factors, it would also be useful to know if changes in mitochondrial dynamics and consequent insulin secretory effects are linked with some or all of these subtypes of the disease in humans.

Do the authors have any comment on targeting mitochondrial dynamics as a therapy in diabetes or if any studies have investigated this either in cell models. mice or human cells. And/or does reversing the negative effects of mitochondrial dynamics recover insulin secretory impairment or content.  

Author Response

We thank the reviewer for the valuable and constructive comments and we addressed them point-by-point in the attached PDF file.

Reviewer 2 Report

In this manuscript, "Mitochondrial Dynamics and Insulin Secretion", Kabra and Jastroch have summarized how mitochondrial dynamics is linked to Insulin Secretion. The topic is interesting and relevant and the manuscript is well written. However, I have some comments and suggestions-

1. The figures should be referenced in the text.

2. The authors describe in detail the Triggering pathway for GSIS, but the Amplification pathway is not well described. I suggest explaining the Amplification pathway properly and incorporating these additional details in Figure 1.

3. Mitophagy and mitochondrial dynamics are inter-related. It is recommended that the authors include a section about mitophagy and GSIS.

4. Lines 145-147 "However, treatment with Troglitazone prevented mitochondrial alteration, loss of β cells, and henceforth impairment in GSIS [37]". Please describe in detail how this drug functions (did this drug target mitochondrial fission/fusion etc?)

5. Figure 2 is not very informative and can be eliminated or combined with Figure 3.

6. I suggest to include diagrams depicting the different domains of  DRP1 and how DRP1 interacts with other proteins (Mff etc) to induce mitochondrial fission. Similarly, include a figure about the fusion proteins OPA1 and MFN1/2 and how these proteins regulate mitochondrial fusion.

7. Line 214- Please resolve the abbreviation "POMC" neurons.

8. The authors nicely summarize the various post-translational modifications on Drp1. It will be informative to add a table summarizing the effects of different post-translational modifications on DRP1 function and how these are relevant in disease context including in T2D.

9. The authors should include in the text a section that highlights the therapeutic potential of targeting mitochondrial dynamics in T2D.

10. Minor issue - There are several sentences in the text that lacks appropriate spacing. The text should be properly formatted.

Author Response

(The authors gave the same response as above.)
